# A framework for studying synaptic plasticity with neural spike train data

**Scott W. Linderman**
Harvard University
Cambridge, MA 02138
swl@seas.harvard.edu

**Christopher H. Stock**
Harvard College
Cambridge, MA 02138
cstock@post.harvard.edu

**Ryan P. Adams**
Harvard University
Cambridge, MA 02138
rpa@seas.harvard.edu

## Abstract

Learning and memory in the brain are implemented by complex, time-varying changes in neural circuitry. The computational rules according to which synaptic weights change over time are the subject of much research, and are not precisely understood. Until recently, limitations in experimental methods have made it challenging to test hypotheses about synaptic plasticity on a large scale. However, as such data become available and these barriers are lifted, it becomes necessary to develop analysis techniques to validate plasticity models. Here, we present a highly extensible framework for modeling arbitrary synaptic plasticity rules on spike train data in populations of interconnected neurons. We treat synaptic weights as a (potentially nonlinear) dynamical system embedded in a fully-Bayesian generalized linear model (GLM). In addition, we provide an algorithm for inferring synaptic weight trajectories alongside the parameters of the GLM and of the learning rules. Using this method, we perform model comparison of two proposed variants of the well-known spike-timing-dependent plasticity (STDP) rule, where nonlinear effects play a substantial role. On synthetic data generated from the biophysical simulator NEURON, we show that we can recover the weight trajectories, the pattern of connectivity, and the underlying learning rules.

## 1 Introduction

Synaptic plasticity is believed to be the fundamental building block of learning and memory in the brain. Its study is of crucial importance to understanding the activity and function of neural circuits. With innovations in neural recording technology providing access to the simultaneous activity of increasingly large populations of neurons, statistical models are promising tools for formulating and testing hypotheses about the dynamics of synaptic connectivity. Advances in optical techniques [1, 2], for example, have made it possible to simultaneously record from and stimulate large populations of synaptically connected neurons. Armed with statistical tools capable of inferring time-varying synaptic connectivity, neuroscientists could test competing models of synaptic plasticity, discover new learning rules at the monosynaptic and network level, investigate the effects of disease on synaptic plasticity, and potentially design stimuli to modify neural networks.

Despite the popularity of GLMs for spike data, relatively little work has attempted to model the time-varying nature of neural interactions. Here we model interaction weights as a dynamical system governed by parametric synaptic plasticity rules. To perform inference in this model, we use particle Markov Chain Monte Carlo (pMCMC) [3], a recently developed inference technique for complex time series. We use this new modeling framework to examine the problem of using recorded data to distinguish between proposed variants of spike-timing-dependent plasticity (STDP) learning rules.

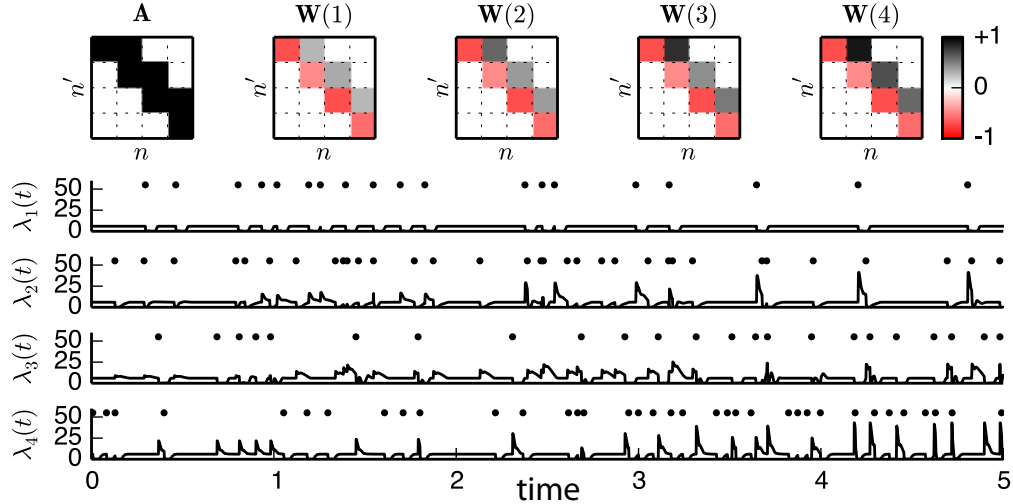

Figure 1: A simple network of four sparsely connected neurons whose synaptic weights are changing over time. Here, the neurons have inhibitory self connections to mimic refractory effects, and are connected via a chain of excitatory synapses, as indicated by the nonzero entries $A_{1\to2}$, $A_{2\to3}$, and $A_{3\to4}$. The corresponding weights of these synapses are strengthening over time (darker entries in $\boldsymbol{W}$), leading to larger impulse responses in the firing rates and a greater number of induced post-synaptic spikes (black dots), as shown below.

## 2 Related Work

The GLM is a probabilistic model that considers spike trains to be realizations from a point process with conditional rate $\lambda(t)$ [4, 5]. From a biophysical perspective, we interpret this rate as a nonlinear function of the cell's membrane potential. When the membrane potential exceeds the spiking threshold potential of the cell, $\lambda(t)$ rises to reflect the rate of the cell's spiking, and when the membrane potential decreases below the spiking threshold, $\lambda(t)$ decays to zero. The membrane potential is modeled as the sum of three terms: a linear function of the stimulus, $\boldsymbol{I}(t)$, for example a low-pass filtered input current, the sum of excitatory and inhibitory PSPs induced by presynaptic neurons, and a constant background rate. In a network of $N$ neurons, let $\mathcal{S}_n = \{s_{n,m}\}_{m=1}^{M_n} \subset [0, T]$ be the set of observed spike times for neuron $n$, where $T$ is the duration of the recording and $M_n$ is the number of spikes. The conditional firing rate of a neuron $n$ can be written,

$$\lambda_n(t) = g\left(b_n + \int_0^t \boldsymbol{k}_n(t-\tau) \cdot \boldsymbol{I}(\tau)\,\mathrm{d}\tau + \sum_{n'=1}^{N}\sum_{m=1}^{M_{n'}} h_{n'\to n}(t - s_{n',m}) \cdot \mathbb{I}[s_{n',m} < t]\right), \quad (1)$$

where $b_n$ is the background rate, the second term is a convolution of the (potentially vector-valued) stimulus with a linear stimulus filter, $\boldsymbol{k}_n(\Delta t)$, and the third is a linear summation of impulse responses, $h_{n'\to n}(\Delta t)$, which preceding spikes on neuron $n'$ induce on the membrane potential of neuron $n$. Finally, the rectifying nonlinearity $g : \mathbb{R} \to \mathbb{R}_+$ converts this linear function of stimulus and spike history into a nonnegative rate. While the spiking threshold potential is not explicitly modeled in this framework, it is implicitly inferred in the amplitude of the impulse responses.

From this semi-biophysical perspective it is clear that one shortcoming of the standard GLM is that it does not account for time-varying connectivity, despite decades of research showing that changes in synaptic weight occur over a variety of time scales and are the basis of many fundamental cognitive processes. This absence is due, in part, to the fact that this direct biophysical interpretation is not warranted in most traditional experimental regimes, e.g., in multi-electrode array (MEA) recordings where electrodes are relatively far apart. However, as high resolution optical recordings grow in popularity, this assumption must be revisited; this is a central motivation for the present model.

There have been a few efforts to incorporate dynamics into the GLM. Stevenson and Koerding [6] extended the GLM to take inter-spike intervals as a covariates and formulated a generalized bilinear model for weights. Eldawlatly et al. [7] modeled the time-varying parameters of a GLM using a dynamic Bayesian network (DBN). However, neither of these approaches accommodate the breadth of synaptic plasticity rules present in the literature. For example, parametric STDP models with hard

bounds on the synaptic weight are not congruent with the convex optimization techniques used by [6], nor are they naturally expressed in a DBN. Here we model time-varying synaptic weights as a potentially nonlinear dynamical system and perform inference using particle MCMC.

Nonstationary, or time-varying, models of synaptic weights have also been studied outside the context of GLMs. For example, Petreska et al. [8] applied hidden switching linear dynamical systems models to neural recordings. This approach has many merits, especially in traditional MEA recordings where synaptic connections are less likely and nonlinear dynamics are not necessarily warranted. Outside the realm of computational neuroscience and spike train analysis, there exist a number of dynamic statistical models, such as West et al. [9], which explored dynamic generalized linear models. However, the types of models we are interested in for studying synaptic plasticity are characterized by domain-specific transition models and sparsity structure, and until recently, the tools for effectively performing inference in these models have been limited.

## 3  A Sparse Time-Varying Generalized Linear Model

In order to capture the time-varying nature of synaptic weights, we extend the standard GLM by first factoring the impulse responses in the firing rate of Equation 1 into a product of three terms:

$$h_{n' \to n}(\Delta t, t) \equiv A_{n' \to n} \, W_{n' \to n}(t) \, r_{n' \to n}(\Delta t). \tag{2}$$

Here, $A_{n' \to n} \in \{0, 1\}$ is a binary random variable indicating the presence of a direct synapse from neuron $n'$ to neuron $n$, $W_{n' \to n}(t) : [0, T] \to \mathbb{R}$ is a non stationary synaptic "weight" trajectory associated with the synapse, and $r_{n' \to n}(\Delta t)$ is a nonnegative, normalized impulse response, i.e. $\int_0^\infty r_{n' \to n}(\tau) \mathrm{d}\tau = 1$. Requiring $r_{n' \to n}(\Delta t)$ to be normalized gives meaning to the synaptic weights: otherwise $W$ would only be defined up to a scaling factor. For simplicity, we assume $r(\Delta t)$ does not change over time, that is, only the amplitude and not the duration of the PSPs are time-varying. This restriction could be adapted in future work.

As is often done in GLMs, we model the normalized impulse responses as a linear combination of basis functions. In order to enforce the normalization of $r(\cdot)$, however, we use a *convex* combination of normalized, nonnegative basis functions. That is,

$$r_{n' \to n}(\Delta t) \equiv \sum_{b=1}^{B} \beta_b^{(n' \to n)} \, r_b(\Delta t),$$

where $\int_0^\infty r_b(\tau) \, \mathrm{d}\tau = 1$, $\forall b$ and $\sum_{b=1}^{B} \beta_b^{(n' \to n)} = 1$, $\forall n, n'$. The same approach is used to model the stimulus filters, $\boldsymbol{k}_n(\Delta t)$, but without the normalization and non-negativity constraints.

The binary random variables $A_{n' \to n}$, which can be collected into an $N \times N$ binary matrix $\boldsymbol{A}$, model the connectivity of the synaptic network. Similarly, the collection of weight trajectories $\{\{W_{n' \to n}(t)\}\}_{n', n}$, which we will collectively refer to as $\boldsymbol{W}(t)$, model the time-varying synaptic weights. This factorization is often called a *spike-and-slab* prior [10], and it allows us to separate our prior beliefs about the structure of the synaptic network from those about the evolution of synaptic weights. For example, in the most general case we might leverage a variety of random network models [11] as prior distributions for $\boldsymbol{A}$, but here we limit ourselves to the simplest network model, the Erdős-Renyi model. Under this model, each $A_{n' \to n}$ is an independent identically distributed Bernoulli random variable with sparsity parameter $\rho$.

Figure 1 illustrates how the adjacency matrix and the time-varying weights are integrated into the GLM. Here, a four-neuron network is connected via a chain of excitatory synapses, and the synapses strengthen over time due to an STDP rule. This is evidenced by the increasing amplitude of the impulse responses in the firing rates. With larger synaptic weights comes an increased probability of postsynaptic spikes, shown as black dots in the figure. In order to model the dynamics of the time-varying synaptic weights, we turn to a rich literature on synaptic plasticity and learning rules.

### 3.1  Learning rules for time-varying synaptic weights

Decades of research on synapses and learning rules have yielded a plethora of models for the evolution of synaptic weights [12]. In most cases, this evolution can be written as a dynamical system,

$$\frac{\mathrm{d}\boldsymbol{W}(t)}{\mathrm{d}t} = \ell\left(\boldsymbol{W}(t), \{s_{n,m} : s_{n,m} < t\}\right) + \epsilon(\boldsymbol{W}(t), t),$$

where $\ell$ is a potentially nonlinear *learning rule* that determines how synaptic weights change as a function of previous spiking. This framework encompasses rate-based rules such as the Oja rule [13] and timing-based rules such as STDP and its variants. The additive noise, $\epsilon(\boldsymbol{W}(t), t)$, need not be Gaussian, and many models require truncated noise distributions.

Following biological intuition, many common learning rules factor into a product of simpler functions. For example, STDP (defined below) updates each synapse independently such that $\mathrm{d}W_{n' \to n}(t)/\mathrm{d}t$ only depends on $W_{n' \to n}(t)$ and the presynaptic spike history $\mathcal{S}_{n<t} = \{s_{n,m} : s_{n,m} < t\}$. Biologically speaking, this means that plasticity is local to the synapse. More sophisticated rules allow dependencies among the columns of $\boldsymbol{W}$. For example, the incoming weights to neuron $n$ may depend upon one another through normalization, as in the Oja rule [13], which scales synapse strength according to the total strength of incoming synapses.

Extensive research in the last fifteen years has identified the relative spike timing between the pre- and postsynaptic neurons as a key component of synaptic plasticity, among other factors such as mean firing rate and dendritic depolarization [14]. STDP is therefore one of the most prominent learning rules in the literature today, with a number of proposed variants based on cell type and biological plausibility. In the experiments to follow, we will make use of two of these proposed variants. First, consider the canonical STDP rule with a "double-exponential" function parameterized by $\tau_-$, $\tau_+$, $A_-$, and $A_+$ [15], in which the effect of a given pair of pre-synaptic and post-synaptic spikes on a weight may be written:

$$\ell\left(W_{n' \to n}(t), \mathcal{S}_{n'}, \mathcal{S}_n\right) = \mathbb{I}[t \in \mathcal{S}_n] \, \ell_+(\mathcal{S}_{n'}; A_+, \tau_+) \; - \; \mathbb{I}[t \in \mathcal{S}_{n'}] \, \ell_-(\mathcal{S}_n; A_-, \tau_-), \quad (3)$$

$$\ell_+(\mathcal{S}_{n'}; A_+, \tau_+) = \sum_{s_{n',m} \in \mathcal{S}_{n'<t}} A_+ \, e^{(t-s_{n',m})/\tau_+} \qquad \ell_-(\mathcal{S}_n; A_-, \tau_-) = \sum_{s_{n,m} \in \mathcal{S}_{n<t}} A_- \, e^{(t-s_{n,m})/\tau_-}.$$

This rule states that weight changes only occur at the time of pre- or post-synaptic spikes, and that the magnitude of the change is a nonlinear function of interspike intervals.

A slightly more complicated model known as the multiplicative STDP rule extends this by bounding the weights above and below by $W_{\mathsf{max}}$ and $W_{\mathsf{min}}$, respectively [16]. Then, the magnitude of the weight update is scaled by the distance from the threshold:

$$\ell\left(W_{n' \to n}(t), \mathcal{S}_{n'}, \mathcal{S}_n\right) = \mathbb{I}[t \in \mathcal{S}_n] \, \tilde{\ell}_+(\mathcal{S}_{n'}; A_+, \tau_+) \, (W_{\mathsf{max}} - W_{n' \to n}(t)),$$

$$- \, \mathbb{I}[t \in \mathcal{S}_{n'}] \, \tilde{\ell}_-(\mathcal{S}_n; A_-, \tau_-) \, (W_{n' \to n}(t) - W_{\mathsf{min}}). \quad (4)$$

Here, by setting $\tilde{\ell}_\pm = \min(\ell_\pm, 1)$, we enforce that the synaptic weights always fall within $[W_{\mathsf{min}}, W_{\mathsf{max}}]$. With this rule, it often makes sense to set $W_{\mathsf{min}}$ to zero.

Similarly, we can construct an additive, bounded model which is identical to the standard additive STDP model except that weights are thresholded at a minimum and maximum value. In this model, the weight never exceeds its set lower and upper bounds, but unlike the multiplicative STDP rule, the proposed weight update is independent of the current weight except at the boundaries. Likewise, whereas with the canonical STDP model it is sensible to use Gaussian noise for $\epsilon(t)$ in the bounded multiplicative model we use truncated Gaussian noise to respect the hard upper and lower bounds on the weights. Note that this noise is dependent upon the current weight, $W_{n' \to n}(t)$.

The nonlinear nature of this rule, which arises from the multiplicative interactions among the parameters, $\theta_\ell = \{A_+, \tau_+, A_-, \tau_-, W_{\mathsf{max}}, W_{\mathsf{max}}\}$, combined with the potentially non-Gaussian noise models, pose substantial challenges for inference. However, the computational cost of these detailed models is counterbalanced by dramatic expansions in the flexibility of the model and the incorporation of *a priori* knowledge of synaptic plasticity. These learning models can be interpreted as strong regularizers of models that would otherwise be highly underdetermined, as there are $N^2$ weight trajectories and only $N$ spike trains. In the next section we will leverage powerful new techniques for Bayesian inference in order to capitalize on these expressive models of synaptic plasticity.

## 4  Inference via particle MCMC

The traditional approach to inference in the standard GLM is penalized maximum likelihood estimation. The log likelihood of a single conditional Poisson process is well known to be,

$$\mathcal{L}\left(\lambda_n(t); \{\mathcal{S}_n\}_{n=1}^N, \boldsymbol{I}(t)\right) = -\int_0^T \lambda_n(t) \, \mathrm{d}t + \sum_{m=1}^{M_n} \log\left(\lambda_n(s_{n,m})\right), \quad (5)$$

and the log likelihood of a population of non-interacting spike trains is simply the sum of each of the log likelihoods for each neuron. The likelihood depends upon the parameters $\theta_{\mathsf{GLM}} = \{b_n, \boldsymbol{k}_n, \{h_{n' \to n}(\Delta t)\}_{n'=1}^N\}$ through the definition of the rate function given in Equation 1. For some link functions $g$, the log likelihood is a concave function of $\theta_{\mathsf{GLM}}$, and the MLE can be found using efficient optimization techniques. Certain dynamical models, namely linear Gaussian latent state space models, also support efficient inference via point process filtering techniques [17].

Due to the potentially nonlinear and non-Gaussian nature of STDP, these existing techniques are not applicable here. Instead we use particle MCMC [3], a powerful technique for inference in time series. Particle MCMC samples the posterior distribution over weight trajectories, $\boldsymbol{W}(t)$, the adjacency matrix $\boldsymbol{A}$, and the model parameters $\theta_{\mathsf{GLM}}$ and $\theta_\ell$, given the observed spike trains, by combining particle filtering with MCMC. We represent the conditional distribution over weight trajectories with a set of discrete particles. Let the instantaneous weights at (discretized) time $t$ be represented by a set of $P$ particles, $\{\boldsymbol{W}_t^{(p)}\}_{p=1}^P$. The particles live in $\mathbb{R}^{N \times N}$ and are assigned normalized *particle weights*[1], $\omega_p$, which approximate the true distribution via $\Pr(\boldsymbol{W}_t) \approx \sum_{p=1}^P \omega_p \, \delta_{\boldsymbol{W}_t^{(p)}}(\boldsymbol{W}_t)$. Particle filtering is a method of inferring a distribution over weight trajectories by iteratively propagating forward in time and reweighting according to how well the new samples explain the data. For each particle $\boldsymbol{W}_t^{(p)}$ at time $t$, we propagate forward one time step using the learning rule to obtain a particle $\boldsymbol{W}_{t+1}^{(p)}$. Then, using Equation 5, we evaluate the log likelihood of the spikes that occurred in the window $[t, t+1)$ and update the weights. Since some of these particles may have very low weights, after each step we resample the particles. After the $T$-th time step we are left with a set of weight trajectories $\{(\boldsymbol{W}_0^{(p)}, \ldots, \boldsymbol{W}_T^{(p)})\}_{p=1}^P$, each associated with a particle weight $\omega_p$.

Particle filtering only yields a distribution over weight trajectories, and implicitly assumes that the other parameters have been specified. Particle MCMC provides a broader inference algorithm for both weights and other parameters. The idea is to interleave *conditional* particle filtering steps that sample the weight trajectory given the current model parameters and the previously sampled weights, with traditional Gibbs updates to sample the model parameters given the current weight trajectory. This combination leaves the stationary distribution of the Markov chain invariant and allows joint inference over weights and parameters. Gibbs updates for the remaining model parameters, including those of the learning rule, are described in the supplementary material.

**Collapsed sampling of $\boldsymbol{A}$ and $\boldsymbol{W}(t)$**   In addition to sampling of weight trajectories and model parameters, particle MCMC approximates the marginal likelihood of entries in the adjacency matrix, $\boldsymbol{A}$, integrating out the corresponding weight trajectory. We have, up to a constant,

$$\Pr(A_{n' \to n} \mid S, \theta_\ell, \theta_{\mathsf{GLM}}, \boldsymbol{A}_{\neg n' \to n}, \boldsymbol{W}_{\neg n' \to n}(t))$$

$$= \int_0^T \int_{-\infty}^\infty p(A_{n' \to n}, W_{n' \to n}(t) \mid S, \theta_\ell, \theta_{\mathsf{GLM}}, \boldsymbol{A}_{\neg n' \to n}, \boldsymbol{W}_{\neg n' \to n}(t)) \, \mathrm{d}W_{n' \to n}(t) \, \mathrm{d}t$$

$$\approx \left[ \prod_{t=1}^T \frac{1}{P} \sum_{p=1}^P \hat{\omega}_t^{(p)} \right] \Pr(A_{n' \to n}),$$

where $\neg n' \to n$ indicates all entries except for $n' \to n$, and the particle weights are obtained by running a particle filter for each assignment of $A_{n' \to n}$. This allows us to jointly sample $A_{n \to n'}$ and $W_{n \to n'}(t)$ by first sampling $A_{n \to n'}$ and then $W_{n \to n'}(t)$ given $A_{n \to n'}$. By marginalizing out the weight trajectory, our algorithm is able to explore the space of adjacency matrices more efficiently.

We capitalize on a number of other opportunities for computational efficiency as well. For example, if the learning rule factors into independent updates for each $W_{n' \to n}(t)$, then we can update each synapse's weight trajectory separately and reduce the particles to one-dimensional objects. In our implementation, we also make use of a pMCMC variant with ancestor sampling [18] that significantly improves convergence. Any distribution may be used to propagate the particles forward; using the learning rule is simply the easiest to implement and understand. We have omitted a number of details in this description; for a thorough overview of particle MCMC, the reader should consult [3, 18].

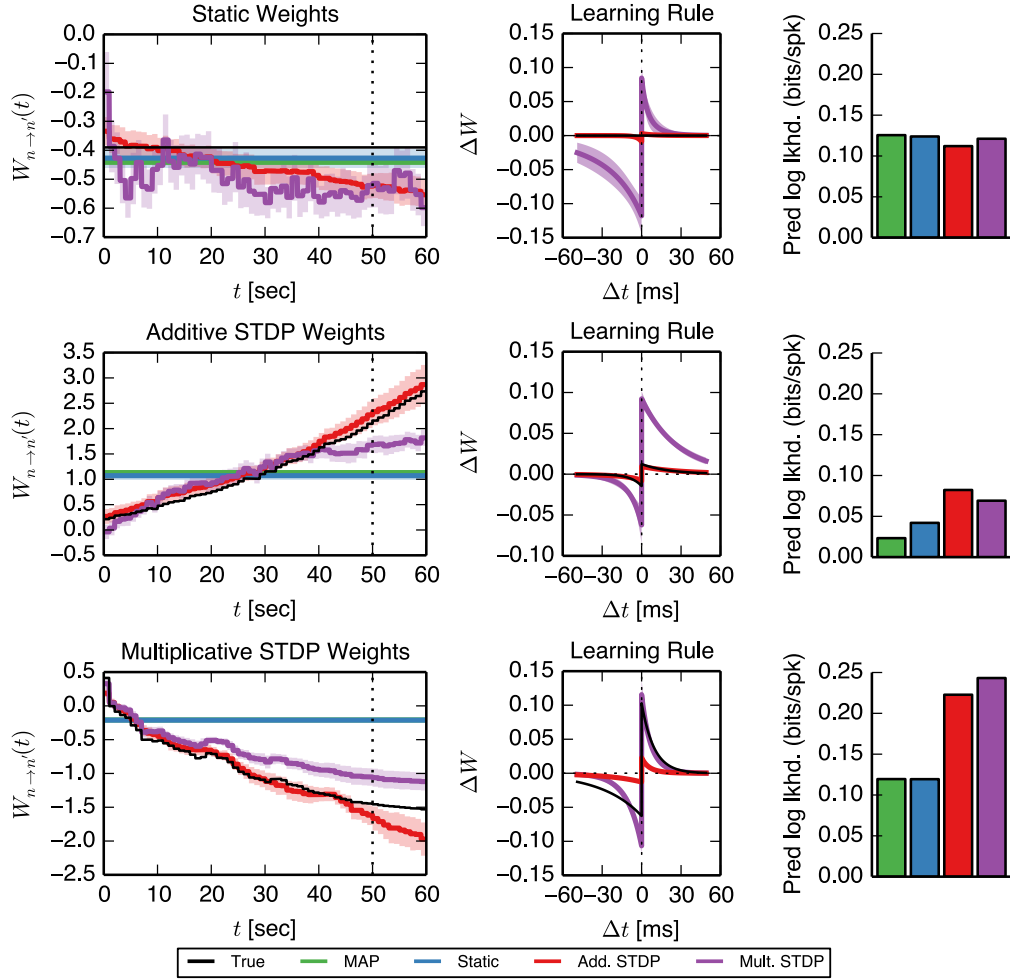

Figure 2: We fit time-varying weight trajectories to spike trains simulated from a GLM with two neurons undergoing no plasticity (top row), an additive, unbounded STDP rule (middle), and a multiplicative, saturating STDP rule (bottom row). We fit the first 50 seconds with four different models: MAP for an L1-regularized GLM, and fully-Bayesian inference for a static, additive STDP, and multiplicative STDP learning rules. In all cases, the correct models yield the highest predictive log likelihood on the final 10 seconds of the dataset.

## 5    Evaluation

We evaluated our technique with two types of synthetic data. First, we generated data from our model, with known ground-truth. Second, we used the well-known simulator NEURON to simulate driven, interconnected populations of neurons undergoing synaptic plasticity. For comparison, we show how the sparse, time-varying GLM compares to a standard GLM with a group LASSO prior on the impulse response coefficients for which we can perform efficient MAP estimation.

### 5.1    GLM-based simulations

As a proof of concept, we study a single synapse undergoing a variety of synaptic plasticity rules and generating spikes according to a GLM. The neurons also have inhibitory self-connections to mimic refractory effects. We tested three synaptic plasticity mechanisms: a static synapse (i.e., no plasticity), the unbounded, additive STDP rule given by Equation 3, and the bounded, multiplicative STDP rule given by Equation 4. For each learning rule, we simulated 60 seconds of spiking activity at 1kHz temporal resolution, updating the synaptic weights every 1s. The baseline firing rates were normally distributed with mean 20Hz and standard deviation of 5Hz. Correlations in the spike timing led to changes in the synaptic weight trajectories that we could detect with our inference algorithm.

Figure 2 shows the true and inferred weight trajectories, the inferred learning rules, and the predictive log likelihood on ten seconds of held out data for each of the three ground truth learning rules. When the underlying weights are static (top row), MAP estimation and static learning rules do an excellent

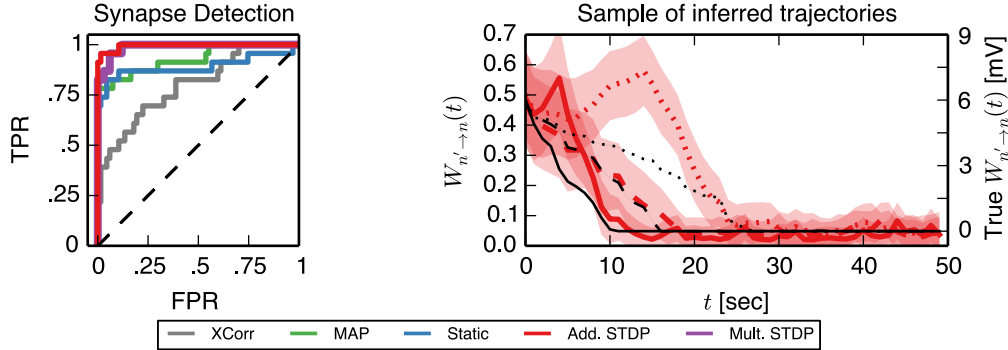

Figure 3: Evaluation of synapse detection on a 60 second spike train from a network of 10 neurons undergoing synaptic plasticity with a saturating, additive STDP rule, simulated with NEURON. The sparse, time-varying GLM with an additive rule outperforms the fully-Bayesian model with static weights, MAP estimation with L1 regularization, and simple thresholding of the cross-correlation matrix.

job of detecting the true weight whereas the two time-varying models must compensate by either setting the learning rule as close to zero as possible, as the additive STDP does, or setting the threshold such that the weight trajectory is nearly constant, as the multiplicative model does. Note that the scales of the additive and multiplicative learning rules are not directly comparable since the weight updates in the multiplicative case are modulated by how close the weight is to the threshold. When the underlying weights vary (middle and bottom rows), the static models must compromise with an intermediate weight. Though the STDP models are both able to capture the qualitative trends, the correct model yields a better fit and better predictive power in both cases.

In terms of computational cost, our approach is clearly more expensive than alternative approaches based on MAP estimation or MLE. We developed a parallel implementation of our algorithm to capitalize on conditional independencies across neurons, i.e. for the additive and multiplicative STDP rules we can sample the weights $W_{*\to n}$ independently of the weights $W_{*\to n'}$. On the two neuron examples we achieve upward of 2 iterations per second (sampling all variables in the model), and we run our model for 1000 iterations. Convergence of the Markov chain is assessed by analyzing the log posterior of the samples, and typically stabilizes after a few hundred iterations. As we scale to networks of ten neurons, our running time quickly increases to roughly 20 seconds per iteration, which is mostly dominated by slice sampling the learning rule parameters. In order to evaluate the conditional probability of a learning rule parameter, we need to sample the weight trajectories for each synapse. Though these running times are nontrivial, they are not prohibitive for networks that are realistically obtainable for optical study of synaptic plasticity.

## 5.2 Biophysical simulations

Using the biophysical simulator NEURON, we performed two experiments. First, we considered a network of 10 sparsely interconnected neurons (28 excitatory synapses) undergoing synaptic plasticity according to an additive STDP rule. Each neuron was driven independently by a hidden population of 13 excitatory neurons and 5 inhibitory neurons connected to the visible neuron with probability 0.8 and fixed synaptic weights averaging 3.0 mV. The visible synapses were initialized close to 6.0 mV and allowed to vary between 0.0 and 10.5 mV. The synaptic delay was fixed at 1.0 ms for all synapses. This yielded a mean firing rate of 10 Hz among visible neurons. Synaptic weights were recorded every 1.0 ms. These parameters were chosen to demonstrate interesting variations in synaptic strength, and as we transition to biological applications it will be necessary to evaluate the sensitivity of the model to these parameters and the appropriate regimes for the circuits under study.

We began by investigating whether the model is able to accurately identify synapses from spikes, or whether it is confounded by spurious correlations. Figure 3 shows that our approach identifies the 28 excitatory synapses in our network, as measured by ROC curve (Add. STDP AUC=0.99), and outperforms static models and cross-correlation. In the sparse, time-varying GLM, the probability of an edge is measured by the mean of $A$ under the posterior, whereas in the standard GLM with MAP estimation, the likelihood of an edge is measured by area under the impulse response.

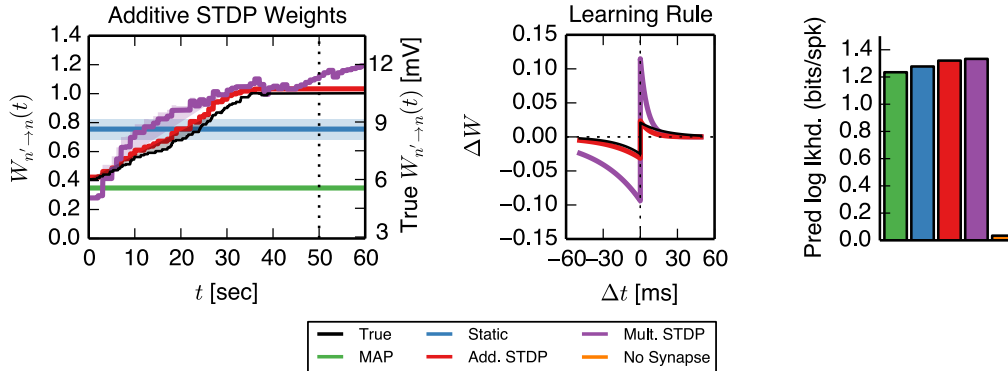

Figure 4: Analogously to Figure 2, a sparse, time-varying GLM can capture the weight trajectories and learning rules from spike trains simulated by NEURON. Here an excitatory synapse undergoes additive STDP with a hard upper bound on the excitatory postsynaptic current. The weight trajectory inferred by our model with the same parametric form of the learning rule matches almost exactly, whereas the static models must compromise in order to capture early and late stages of the data, and the multiplicative weight exhibits qualitatively different trajectories. Nevertheless, in terms of predictive log likelihood, we do not have enough information to correctly determine the underlying learning rule. Potential solutions are discussed in the main text.

Looking into the synapses that are detected by the time-varying model and missed by the static model, we find an interesting pattern. The improved performance comes from synapses that decay in strength over the recording period. Three examples of these synaptic weight trajectories are shown in the right panel of Figure 3. The time-varying model assigns over 90% probability to each of the three synapses, whereas the static model infers less than a 40% probability for each synapse.

Finally, we investigated our model's ability to distinguish various learning rules by looking at a single synapse, analogous to the experiment performed on data from the GLM. Figure 4 shows the results of a weight trajectory for a synapse under additive STDP with a strict threshold on the excitatory postsynaptic current. The time-varying GLM with an additive model captures the same trajectory, as shown in the left panel. The GLM weights have been linearly rescaled to align with the true weights, which are measured in millivolts. Furthermore, the inferred additive STDP learning rule, in particular the time constants and relative amplitudes, perfectly match the true learning rule.

These results demonstrate that a sparse, time-varying GLM is capable of discovering synaptic weight trajectories, but in terms of predictive likelihood, we still have insufficient evidence to distinguish additive and multiplicative STDP rules. By the end of the training period, the weights have saturated at a level that almost surely induces postsynaptic spikes. At this point, we cannot distinguish two learning rules which have both reached saturation. This motivates further studies that leverage this probabilistic model in an optimal experimental design framework, similar to recent work by Shababo et al. [19], in order to conclusively test hypotheses about synaptic plasticity.

# 6 Discussion

Motivated by the advent of optical tools for interrogating networks of synaptically connected neurons, which make it possible to study synaptic plasticity in novel ways, we have extended the GLM to model a sparse, time-varying synaptic network, and introduced a fully-Bayesian inference algorithm built upon particle MCMC. Our initial results suggest that it is possible to infer weight trajectories for a variety of biologically plausible learning rules.

A number of interesting questions remain as we look to apply these methods to biological recordings. We have assumed access to precise spike times, though extracting spike times from optical recordings poses inferential challenges of its own. Solutions like those of Vogelstein et al. [20] could be incorporated into our probabilistic model. Computationally, particle MCMC could be replaced with stochastic EM to achieve improved performance [18], and optimal experimental design could aid in the exploration of stimuli to distinguish between learning rules. Beyond these direct extensions, this work opens up potential to infer latent state spaces with potentially nonlinear dynamics and non-Gaussian noise, and to infer learning rules at the synaptic or even the network level.

**Acknowledgments** This work was partially funded by DARPA YFA N66001-12-1-4219 and NSF IIS-1421780. S.W.L. was supported by an NDSEG fellowship and by the NSF Center for Brains, Minds, and Machines.

## Footnotes

[1]Note that the particle weights are *not* the same as the synaptic weights.

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
