[Supplementary Material]

# A framework for studying synaptic plasticity with neural spike train data
# Supplementary Material

**Scott W. Linderman**
Harvard University
Cambridge, MA 02138
swl@seas.harvard.edu

**Christopher H. Stock**
Harvard College
Cambridge, MA 02138
cstock@post.harvard.edu

**Ryan P. Adams**
Harvard University
Cambridge, MA 02138
rpa@seas.harvard.edu

## 1    Details of the Inference Algorithm

The main text describes the core of the inference algorithm for sampling the weights, $\boldsymbol{W}(t)$, and the adjacency matrix, $\boldsymbol{A}$. There are a number of other parameters that we infer as well, as described here.

**Sampling the impulse responses, $\boldsymbol{r}(\Delta t)$**    Recall that the impulse responses are modeled as,

$$r_{n' \to n}(\Delta t) \equiv \sum_{b=1}^{B} \beta_b^{(n' \to n)} \, r_b(\Delta t),$$

$$\boldsymbol{\beta}^{(n' \to n)} \sim \text{Dirichlet}(\alpha),$$

where $\int_0^\infty r_b(\tau) \, \mathrm{d}\tau = 1$, $\forall b$ and $\alpha$ is the parameter of a symmetric Dirichlet distribution. We sample the impulse response coefficients, $\boldsymbol{\beta}^{(n' \to n)}$, using Hamiltonian Monte Carlo. To avoid boundary constraints, we use the "expanded-mean" parameterization described in Patterson and Teh [1]. Specifically, we let,

$$\beta_b^{(n' \to n)} = \frac{|\theta_b^{(n' \to n)}|}{\sum_{b'=1}^{B} |\theta_{b'}^{(n' \to n)}|},$$

$$|\theta_b^{(n' \to n)}| \sim \text{Gamma}(\alpha, 1).$$

In our simulations we let $\alpha = 1$ and $\Delta t_{\mathsf{max}} = 100\text{ms}$. Our impulse response basis vectors, $r_b(\Delta t)$ consist of $B = 6$ rectified cosine tuning curves, as described in [2].

**Sampling learning rule parameters, $\theta_\ell$**    The learning rules themselves also possess parameters, e.g., the amplitude of the STDP update, $A_+$. One of the benefits of particle MCMC is that each iteration yields samples of the weight trajectories. Given these trajectories, it is generally straightforward to employ Gibbs sampling on the parameters of the learning rule. The conditional probability of $\theta_\ell$ is a function of how much the current weight trajectory differs from that predicted by a learning rule with parameters $\theta_\ell$. We place gamma priors on the nonnegative parameters, $A_+$, $A_-$, $\tau_+$, and $\tau_-$. We use shape parameters $a = 1$ and rate parameters of 50, 150, 100, and 100, respectively (time constants are measured in seconds). We restrict the weight boundaries such that $W_{\mathsf{max}} > 0$ and $W_{\mathsf{min}} < 0$, and place gamma priors on these as well. For the NEURON data, which consists of purely excitatory connections, we set $W_{\mathsf{max}} \sim \text{Gamma}(1, 1)$ and $-W_{\mathsf{min}} \sim \text{Gamma}(1, 100)$.

We sample the conditional distributions using slice sampling. In theory, particle marginal Metropolis-Hastings updates [3] may yield improved convergence, for example when there are strong dependencies between the current weight trajectory and the weight bounds, but in practice we find that slice sampling is sufficient for our purposes.

**Sampling static refractory weights,** $W_{n \to n}$    Though weights between neurons may change as a result of activity, it is less clear that self weights in the GLM, which effectively implement refractoriness, should change. In our simulations, we set a self-inhibitory prior on the self weights, $W_{n \to n}$ $\mathcal{N}(-1.0, 0.5)$. For most typical choices of nonlinearities, $g(\cdot)$, specifically those which are both convex and log concave, the conditional distribution of $W_{n \to n}$ will be log concave if its prior is. This condition is met by a Gaussian prior, and renders the conditional distribution amenable to adaptive rejection sampling (ARS). Furthermore, if we wish to sample the presence or absence of a self connection $A_{n \to n}$, then under a Gaussian prior we may use a joint approach as we do with the time varying weights. Here, the marginal probability of an edge may be approximated using Gauss-Hermite quadrature. Then, the weights may be sampled using ARS, where the abscissae of the quadrature may seed the hull of the conditional distribution.

**Sampling the bias parameters,** $x_0$    Under typical choices of nonlinearity, $g$, and under a log concave prior, the conditional distribution of $x_0^{(n)}$ is log concave and amenable to adaptive rejection sampling. In practice, however, we opt for Hamiltonian Monte Carlo, as with the parameters of the impulse responses.

**Computational details**    Our inference algorithm was implemented in Python and built upon the Theano framework for automatic differentiation and compilation to C or GPU kernels. The code may be found at `https://github.com/slinderman/pyglm`. Though we have opted for a fully-Bayesian approach, a particle SAEM approach could be used instead and may offer substantial improvements in runtime while yielding similar results [4].