[Reviews · NeurIPS 2014]

Submitted by Assigned_Reviewer_7

This paper provides an inference technique for plasticity models. It generalizes traditional GLM to tackle the more ecological situation where synaptic weights changes over time. This paper uses particular MCMC as the work horse for inference and hence inherits its pros and cons. The STDP rule can be flexible, but the cost of long computational time. Experiments show that the inference technique can reliably identify the right STDP rule based on data generated from the model, and show, to some degree, ability to distinguish between STDP rules using data generated from NEURON.

This paper is clearly-written, well-executed, addresses an original problem and has the potential to get many citations. It would receive more citation if it provides an rough estimation of the complexity as a function of neurons/connections.
Summary: The well-written paper tackles the novel and important problem of synaptic plasticity in the analysis of network connectivity.

Submitted by Assigned_Reviewer_8

In this paper, the authors present a beefed up GLM model where the matrix which defines interactions between the neurons is time-varying based on some learning rule such as STDP. Also, sparsity is promoted with a spike-and-slab like factorization of the weight matrix. Inference in the model is performed using particle MCMC which is a powerful hybridization of SMC and MCMC (in this case Gibbs sampling).

I think this paper presents an important variation in models of neural dynamics in that it attempts to model time-varying dynamics - a problem that computational neuroscientists and the NIPS community are likely interested in. While it doesn't assess its performance on real data, it does assess its performance on data generated from a biophysical model with unobserved neurons - the latter being an important component to consider. Though the results are on toy networks, the method's potential is convincing. Also, it is the first application of pMCMC to computational neuroscience that I'm aware of.

That being said, I have one main area of concern in this draft.

I think the relationship between this model and the current methods in experimental neuroscience is not always presented correctly. In line 41 the word "optogenetics" may be inappropriate. Optogenetics is not related to recording activity, only stimulating. Maybe use the word optical here instead of optogenetic. I'd also reconsider the usage of reference 2. That is a voltage imaging paper where the analysis requires trial averaging. Also, voltage imaging is not really ready to be used for population imaging. However, a more recent paper from Adam Cohen's group does have a demonstration of this (All-optical electrophysiology in mammalian neurons using engineered microbial rhodopsins). Generally people use Ca++ indicators for population imaging. For recent advances in Ca++ imaging, see "Ultrasensitive fluorescent proteins for imaging neuronal activity" from a collaboration of Janelia Farm labs. There are plenty of other Ca++ imaging papers that would be appropriate as well. The one I listed doesn't demonstrate poulation imaging, but the indicator from that paper is regularly used to image populations of neurons in vitro and in vivo.

The authors often cite optical methods as motivation for their model. The idea is that optical methods allow the recording of cells that are near each other and presumably some of them are synaptically connected. There is a major issue that goes unmentioned in this paper: extracting spikes from imaging data is a non-trivial problem (see papers by Josh Vogelstein and Liam Paninski for commonly used algorithms), and the temporal resolution of imaging does not often permit the precise measurements needed to identify STDP type rules. The papers referenced which do infer synaptic connections [5,6] use electrophysiological data which does provide highly accurate and precise spike times. That being said, fast and reliable optical indicators of action potentials are likely on the horizon, and the model is still interesting even if it does not admit a causal, synaptic interpretation.

Besides that, there are a few smaller specific points that should be clarified.

Because the GLM model does not necessarily admit a true PSP or PSC interpretation, using units of nA here maybe disingenuous. Furthermore, for PSC amplitudes, even 1 nA would be very big. One might expect a max current in the range of 5-100 of pA. In addition, you reference PSPs earlier (post-synaptic potentials) in the paper. These would be measured in millivolts not pico- or nanoamps. A PSC (post-synaptic current) would be measured in amperes. Because the membrane resistance changes over time, the two are not interchangeable.

Given that the probability of spiking is voltage-dependent, not current-dependent, you could stick with a PSP interpretation and change the units to millivolts. That being said, many STDP models vary the max current parameter - they don't explicitly model some max potential change. Perhaps it's worth considering dropping the units here and saying that the unitless GLM weights are modulated in a way that is based on STDP. I think this is reasonable and avoids the slippery-slope of claiming your model captures electrophysiological properties.

It's not clear how the noise variable is truncated to ensure that the bounds are maintained. Is it a function of W(t) and dW(t) such that you update the weights without noise and then truncate the Gaussian at (0+W(t)+dW(t),W_max-W(t)+dW(t))? Or is it dependent on A_+ and A_-? In any case, it's not clear how the model results in noise from a truncated Gaussian random variable which will not violate the bounds.

The way you've written it, it seems like you are claiming the model in Song et al. is infeasible because the weights are unbounded. But they do in fact bound synaptic weights from below at 0 and from above by some $g_{max}$ (in their notation).

It's not clear to me that Eldawlatly et al. model time-varying synaptic parameters - as you claim - as much as causal influences that act at multiple time lags in a discrete-time model. The interpretation - depending on the bin size and length of the delay - could be either indirect (e.g. polysynaptic effects) or direct (e.g. delay due to axon length), but none of the model parameters are time-varying.

Finally, I'm concerned about the tractability of the inference algorithm. It is not uncommon for optical recordings to contain 100s of neurons. How long would a network of that size take to fit? How much data would be needed to fit the model?
Summary: While there are some issues in terms of clarity and how the authors place their model into the framework of experimental neuroscience, I think this paper is a novel contribution and will be of interst to the NIPS community.

Submitted by Assigned_Reviewer_10

The authors describe a model for dynamic synaptic weights embedded within a more typical GLM of multiple, interconnected neurons. They show how the weights of these synapses can be inferred over time, and thus how learning rules for real neuronal systems can be uncovered. The authors demonstrate the efficacy of such an inference scheme on modeled data (NEURON), for which they know the ground truth.

The authors underline the significance of this work by pointing the fact that whereas most classic multi-neuron studies have used electrode arrays with wide spacing (and little synaptic connectivity), modern experiments routinely capture dense neuronal populations with high interconnectivity. These recordings are likely to contain signals related to synaptic plasticity, and network models that can explain such phenomena are of great interest.

The manuscript lays out such a model in a clear and elegant fashion. The inference method appears solid. I have no major qualms with the manuscript and think that it would be a good addition to the NIPS conference agenda.
Summary: This manuscript presents a novel and elegant model that could be a useful tool for experimentalists.
Author Feedback
Author rebuttal: We thank the reviewers for their thoughtful and constructive feedback. Your suggestions will certainly help us strengthen and clarify the paper.

We would like to address some of Reviewer 3's comments in particular. We will rephrase our motivation in terms of "optical" methods, and will also cite the recent developments in fast Ca++ indicators. You also bring up a good point about spike extraction from imaging data - in addition to citing Vogelstein and Paninski, we will point out that spike extraction could be incorporated into this probabilistic model. We will also address the smaller points to elucidate the related works and truncated noise. We appreciated your comments on the biological interpretation of our model, and we particularly like your suggestion of measuring synaptic weights in units of millivolts. Since we are treating the pre-nonlinearity activation as analogous to the membrane potential, adopting millivolts is a natural choice.

Finally, we agree that a more thorough discussion of computational concerns and limitations will be valued by the NIPS community. We are happy to run additional experiments to help clarify and strengthen our results in this respect, and for now we will note that many of the variables in our model can be sampled in parallel. Our implementation actually uses IPython parallel and Theano to distribute the inference algorithm on a GPU cluster. This is not necessary for small toy networks, but will be vital as we scale to larger populations of neurons. This code will be made available alongside the paper.